# Biomaterial-Mediated Protein Expression Induced by Peptide-mRNA Nanoparticles Embedded in Lyophilized Collagen Scaffolds

**DOI:** 10.3390/pharmaceutics14081619

**Published:** 2022-08-02

**Authors:** Rik Oude Egberink, Helen M. Zegelaar, Najoua El Boujnouni, Elly M. M. Versteeg, Willeke F. Daamen, Roland Brock

**Affiliations:** 1Department of Biochemistry, Radboud Institute for Molecular Life Sciences (RIMLS), Radboud University Medical Center, Geert Grooteplein 28, 6525 GA Nijmegen, The Netherlands; rik.oudeegberink@radboudumc.nl (R.O.E.); helen.zegelaar@ru.nl (H.M.Z.); najoua.elboujnouni@radboudumc.nl (N.E.B.); elly.versteeg@radboudumc.nl (E.M.M.V.); willeke.daamen@radboudumc.nl (W.F.D.); 2Department of Cell Biology, Radboud Institute for Molecular Life Sciences (RIMLS), Radboud University Medical Center, Geert Grooteplein 28, 6525 GA Nijmegen, The Netherlands; 3Department of Medical Biochemistry, College of Medicine and Medical Sciences, Arabian Gulf University, Manama 329, Bahrain

**Keywords:** biomaterial, bone regeneration, cell-penetrating peptide, collagen scaffold, drug delivery, mRNA, nanomedicine, nanoparticle, polyplexes, regenerative medicine, tissue engineering

## Abstract

In our aging society, the number of patients suffering from poorly healing bone defects increases. Bone morphogenetic proteins (BMPs) are used in the clinic to promote bone regeneration. However, poor control of BMP delivery and thus activity necessitates high doses, resulting in adverse effects and increased costs. It has been demonstrated that messenger RNA (mRNA) provides a superior alternative to protein delivery due to local uptake and prolonged expression restricted to the site of action. Here, we present the development of porous collagen scaffolds incorporating peptide-mRNA nanoparticles (NPs). Nanoparticles were generated by simply mixing aqueous solutions of the cationic cell-penetrating peptide PepFect14 (PF14) and mRNA. Peptide-mRNA complexes were uniformly distributed throughout the scaffolds, and matrices fully preserved cell attachment and viability. There was a clear dependence of protein expression on the incorporated amount of mRNA. Importantly, after lyophilization, the mRNA formulation in the collagen scaffolds retained activity also at 4 °C over two weeks. Overall, our results demonstrate that collagen scaffolds incorporating peptide-mRNA complexes hold promise as off-the-shelf functional biomaterials for applications in regenerative medicine and constitute a viable alternative to lipid-based mRNA formulations.

## 1. Introduction

With the clinical approval of mRNA-based SARS-CoV2 vaccines, the potential of messenger RNA (mRNA) as a therapeutic modality is increasingly being recognized [1,2,3,4]. mRNA is a transient entity that acts as an intermediary between gene and protein. When mRNA is delivered into target cells, it temporarily induces protein expression, followed by mRNA degradation by physiological breakdown mechanisms. Therefore, mRNA delivery is a traceless transfection method [5].

Accomplishing successful mRNA delivery, however, is not a trivial task. Both the size of mRNA molecules and the negative charge of the phosphodiester-backbone hamper the introduction of mRNA into cells. Moreover, systemic administration of mRNA can trigger activation of innate immunity reminiscent of invading pathogens [6]. Even a single change, such as base oxidation or a strand break in the mRNA, can render the mRNA molecule ineffective [7]. Very clearly, there is a strong need to formulate mRNA such that it is protected from local microenvironments and that it achieves efficient cellular uptake while simultaneously mitigating its potential adverse effects that may arise from recognition by innate immune receptors.

Formulation strategies for mRNA are based on the non-covalent, charge-driven complexation of the negatively charged mRNA with a positively charged delivery vehicle. Both polymer-based and lipid-based approaches are being explored. Lipid nanoparticles (LNPs) are the most widely used formulation. For LNP formation, an ethanolic solution of lipids is mixed with an aqueous solution of mRNA using dedicated microfluidic devices [8]. Alternatively, the amphipathic cationic cell-penetrating peptide PepFect14 (PF14) is an example of a polymer-based approach [9]. In contrast to LNPs, peptide-mRNA nanoparticles can be obtained by simply mixing the aqueous solutions. Moreover, their composition is less complex than LNPs as only one type of peptide is needed instead of a mixture of lipids. Incorporating mRNA NPs into biomaterials holds great promise for tissue regeneration, where endogenous cells at the site of injury are instructed to promote the healing of damaged tissues [10,11,12,13]. 

Biomaterials are particularly well-suited for applications in bone regeneration as these can provide a structurally rigid framework for attachment of host cells and accelerate cell growth and differentiation. Bone morphogenetic proteins (BMPs) have been explored extensively as pro-osteogenic growth factors to promote bone regeneration in poorly healing and critical-sized bone defects [14]. For delivery of BMP-2, an implantable metal cage containing the recombinant human BMP-2 (rhBMP-2) protein in an absorbable bovine collagen sponge was FDA-approved for anterior lumbar fusion in 2002 [15]. However, this type of delivery is not without problems. BMPs are soluble proteins, which dissipate from their intended locations, diluting local concentrations and thus reducing potential efficacy [16]. Moreover, very high doses of BMP-2 have been reported to result in tissue inflammation, wound complications, peripheral oedemas, osteolysis, and ectopic bone formation [17,18,19]. Proteinaceous growth factor delivery, without a delivery vehicle or device, provides no control over delivery or activity, as demonstrated by intraosseous infusion of rhBMP-2 in a porcine ischemic osteonecrosis model, where the therapeutic benefit was absent at non-toxic doses. In contrast, at effective doses, heterotopic ossification (i.e., ectopic bone formation) was observed [20,21].

Collagen sponges, however, show burst release, where nearly 50% of the initial rhBMP-2 dose is released within the first 24 h [22]. Contrary to the long-held belief that BMPs should be present at high concentrations for the entire duration of the bone regeneration process, recent results have shown that an initial, transient burst of BMPs can reach a desired therapeutic effect without the adverse effects (such as ectopic callus formation) observed for longer durations of rhBMP-2 delivery [23,24]. 

Additionally, besides BMP-2 other BMPs have been investigated [25,26,27], where BMP-7 showed auspicious characteristics in terms of accelerating bone healing and was associated with fewer complications than traditional bone grafting [28,29,30,31]. However, in a large-scale clinical study, BMP-7 infusion did not show superiority over autografts [32], leading to the rejection of the pre-market approval of BMP-7 in 2009, despite FDA Humanitarian Device Exemption approval in 2004 [33]. As an alternative to rhBMP-2, BMP-2 mRNA LNPs have been incorporated into collagen matrices [13,23,34,35,36,37,38,39]. Messenger RNA yields protein expression for several days without a strong initial burst. BMP-2 mRNA-functionalized collagen scaffolds outperformed proteinaceous delivery by ~50-fold [40]. 

As described, peptide-based mRNA formulations only require one component for delivery and are easier to produce than mRNA-LNPs. We have shown that these nanoparticles yield protein expression in the peritoneal cavity upon intraperitoneal injection [41]. Here, we establish the incorporation of these nanoparticles into porous collagen scaffolds. Using fluorescently labeled mRNA, we demonstrate that peptide-mRNA nanoparticles are uniformly distributed throughout the scaffold. The scaffolds can be lyophilized, and significant mRNA activity is maintained upon storage at 4 °C for at least two weeks. Furthermore, there was a clear dependence of protein expression on mRNA dose. Collectively, our results establish peptide mRNA-functionalized scaffolds as an off-the-shelf storable alternative to LNP mRNA scaffolds.

## 2. Materials and Methods

### 2.1. Cell Lines and Culture Media

Subconfluent cultures of the MC3T3-E1 subclone 4 (CRL-2593, American Type Culture Collection; ATCC, Manassas, VA, USA) pre-osteoblastic murine cell line were maintained in Minimal Essential Medium α (MEM-α; Gibco, Waltham, MA, USA, Catalog number: A10490-01) and C2C12 murine myoblasts (CRL-1772, ATCC) were maintained in Dulbecco’s MEM (DMEM; Gibco, Cat No. 41966-029). Both culture media were supplemented with 10% *v*/*v* fetal bovine serum (FBS; Gibco, Cat No. 10270-106). All experiments were carried out with MC3T3 and C2C12 cells with a passage number lower than 30.

### 2.2. Preparation of Collagen Scaffolds

Porous collagen scaffolds were prepared as previously described [42]. In brief, 0.8% *w*/*v* of collagen type I insoluble fibrils (isolated from bovine Achilles tendon, according to the in-house protocol [43]) were suspended in 0.25 M acetic acid (Sigma-Aldrich, St. Louis, MO, USA) and left to swell overnight at 4 °C. After homogenization and removal of air bubbles, 4 mL of the collagen suspension was poured into each well (9.6 cm^2^) of a 6-well plate (Greiner Cellstar, Frickenhausen, Germany) and frozen for at least 1 h at ≤−20 °C and lyophilized overnight in a ScanVac CoolSafe freeze dryer (Labogene, Lillerød, Denmark). These steps yielded collagen scaffolds with isotropic pores and a thickness of 4 mm [42].

Crosslinks were introduced by vapor fixation of the collagen scaffolds with 105 µL of 37% *w*/*v* formaldehyde (VWR, Radnor, PA, USA) per mg of collagen scaffold for 30 min under vacuum in an 18.5 L desiccator (Duran Group, Duisburg, Germany), followed by quenching with 30 mM NaBH_4_ in 100 mM phosphate buffer pH 6.5 at 4 °C. Afterward, collagen scaffolds were washed six times for 30 min in Milli-Q water (MQ) also at 4 °C. Before lyophilization, scaffolds were placed into custom-made filter dishes. These dishes were prepared by punching a 5 mm hole with a biopsy puncher (VWR, Cat No. LBMI48501) in the bottom of a 60 × 15 mm (diameter × height) round Petri dish, to which 0.2 µm filter paper was firmly taped. The filter dishes were wrapped with parafilm during lyophilization to prevent unintended opening. After lyophilization, collagen scaffolds shrunk to a height of 2 to 3 mm. From these, smaller collagen scaffolds were made using a 12 mm biopsy puncher (Acu-Derm, Fort Lauderdale, FL, USA). Lastly, collagen scaffolds were sterilized by UV-C irradiation using the default sterilization procedure of the GS Gene Linker UV Chamber (Bio-Rad, Hercules, CA, USA) for 15 min, flipped with sterile tweezers, followed by UV irradiation for an additional 15 min.

### 2.3. Messenger RNA (mRNA)

5-methoxyuridine-substituted Cyanine 5 (Cy5)-labeled eGFP mRNA was purchased from Trilink Biotechnologies (L-7701, San Diego, CA, USA). This Cy5-eGFP mRNA has a length of 996 nucleotides (nt), was capped using CleanCap technology, and polyadenylated (276 nt). mRNA coding for secreted nanoluciferase (SecNLuc) and BMP-7 mRNA (NCBI Reference Sequence: NM_001719.3) were both obtained from RiboPro (Oss, The Netherlands). These mRNAs were capped with a Cap 1 structure, depleted of double-stranded RNAs, and sequence-modified to reduce immunogenicity. All mRNAs were snap-frozen in liquid nitrogen and stored at 100 ng µL^−1^ in Milli-Q (MQ) in DNA LoBind tubes (Eppendorf, Hamburg, Germany) at −80 °C until use. Before use, the mRNA solution was thawed and kept on ice.

### 2.4. Formation of Transfection Complexes

Transfection complexes were prepared as described previously [44] using the cell-penetrating peptide PepFect14 (PF14) with the following sequence: Stearyl-AlaGlyTyrLeuLeuGlyLysLeuLeuOrnOrnLeuAlaAlaAlaAlaLeuOrnOrnLeuLeu-NH_2_, where Orn denotes the non-proteinogenic amino acid ornithine and -NH_2_ indicates a C-terminal amidation (purchased from EMC Microcollections, Tübingen, Germany; Appendix A). Nanoparticles with mRNA were formed by a ‘50/50 stream method’. Two stock solutions of mRNA and PF14 were prepared in MQ and simultaneously aspirated with electronically dispensing pipettes (E4 Electronic Pipette, LTS E4-100XLS+, Mettler-Toledo Rainin, LLC, Oakland, CA, USA) at a flow rate of 11 mL min^−1^. The pipette tips were inserted into a custom-made 1.5 mL Eppendorf tube holder to collect the solution. The angle between both pipette tips was 75°, and the angle between the pipette tips and the tube wall was 45°. PF14 polyplexes were formed at ten times the concentration for loading of scaffolds at a nitrogen/phosphate (N/P) ratio of 3. Each µM (final concentration) of PF14 in 200 µL corresponds to a final mRNA quantity of 107.32 ng mRNA/collagen scaffold of 12 mm (so 2.5 µM PF14 equals 268.3 ng mRNA, 5 µM PF14 equals 536.6 ng mRNA).

For the formation of polycationic lipid-based complexes, Lipofectamine MessengerMAX (LMM; Thermo Fisher Scientific, Waltham, MA, USA) was used according to the manufacturer’s instructions. In short, LMM was incubated in Opti-MEM (Gibco, Cat. No. 11058021) for 10 min at room temperature (RT). The appropriate amount of mRNA solution was diluted in Opti-MEM and incubated with LMM for at least 5 min at RT. This procedure yielded a lipoplex mixture with a concentration of 10 ng mRNA µL^−1^. All untreated controls were made by adding the same ratio of MQ to cell culture medium as in the experimental conditions.

The hydrodynamic size of the nanoparticles was measured at 25 °C by dynamic light scattering (DLS) using a Zetasizer Nano ZS (Malvern Instruments, Worcestershire, UK) equipped with a 4 mW He-Ne laser (633 nm) with a backscatter detection angle of 173°. A total of 40 µL of 10× concentrated NP (both PF14 and LMM) solution was measured in a UV-cuvette (BrandTech Scientific, Essex, CT, USA, Cat. No. 759200).

### 2.5. Nanoparticle (NP) Diffusion into Collagen Scaffolds

Collagen scaffolds were loaded with 50 µL of 2 µM PF14 + Cy5-eGFP mRNA at N/P = 3 (corresponding to 53.6 ng/collagen scaffold). mRNA-loaded collagen scaffolds of 12 mm were placed inside a 60 × 15 mm (diameter × height) filtered Petri dish and incubated at 37 °C in a humidified incubator for 1, 4, and 24 h. Subsequently, collagen scaffolds were lyophilized. To investigate the homogeneity of loading, freeze-dried collagen scaffolds were cut into paper-thin sections with a scalpel and subsequently reswollen in 500 µL of MQ. These cross-sections were either cut in the same direction as mRNA NP loading (top) or from the opposite side of mRNA NP loading (bottom), followed by mounting the cross-sections on a 1 mm thick, rectangular 75 × 25 mm SuperFrost Plus glass slides (Menzel Gläser, Braunschweig, Germany).

Cy5-eGFP mRNA signals were visualized using a Leica TCS SP8 SMD (Leica Microsystems, Mannheim, Germany) confocal laser scanning microscope (CLSM) equipped with an HCX PL APO 10×/0.40 dry objective lens. Cy5 was excited with a white-light laser (WLL) at 633 nm, and emission was collected between 650 and 690 nm using a photomultiplier tube (PMT) detector. Z-stacks were made such that the entire imageable depth of the section of the collagen scaffold was visualized. Intensity profiles of Cy5-eGFP mRNA signals across the section were acquired along the entire width of the image in ImageJ (version 1.53f51) by making an 8-bit Z-projection of average intensities.

### 2.6. Optimization of the Cell Seeding Density

Several seeding densities of C2C12 cells were tested for viability inside the 3D collagen scaffolds by dropwise addition of 100 µL complete medium containing 1 × 10^5^, 5 × 10^5^, or 1 × 10^6^ cells per UV-sterilized collagen scaffold of 12 mm. Scaffolds were put into individual wells of a 24-well plate. After cell seeding, scaffolds were incubated in a humidified incubator at 37 °C for 1 h, followed by adding 500 µL of medium and additional incubation for 48 h. For visualization of cell growth, the medium was removed, and collagen scaffolds were washed with phosphate-buffered saline (PBS). Then, 500 µL of 3 µM calcein-acetoxymethyl ester (Calcein-AM, BioLegend, San Diego, CA, USA, Cat No. 425201) and 2 µg mL^−1^ of propidium iodide (PI, Sigma-Aldrich, Cat No. P4170) in PBS were added to the scaffolds and incubated in a 37 °C with 5% CO_2_ humidified incubator for 10 min. 

Directly following the incubation, live-cell imaging was performed using the Leica TCS SP8 SMD with an HCX PL APO 10×/0.40 dry objective lens and a temperature-controlled stage at 36.5 °C. Calcein-AM was excited at 496 nm using a WLL, with emission collected between 505 and 540 nm. The PI signal was sequentially acquired by excitation at 535 nm and emission collection between 605 and 645 nm.

### 2.7. Presence of Viable Cells 24 h Post-Seeding

The penetration depth of viable MC3T3 and C2C12 cells into the scaffolds was assessed 24 h after initial seeding by staining of cells with 1 µM of CellTrace Yellow (Thermo Fisher Scientific, Cat. No. C34567), according to the manufacturer’s instructions. After staining for 20 min in a humidified incubator at 37 °C, 1.5 × 10^6^ cells were seeded per collagen scaffold. After 24 h, collagen scaffolds were fixed with 500 µL of 4% *w*/*v* paraformaldehyde (Merck Millipore, Burlington, MA, USA) in PBS, followed by optical clearing with See Deep Brain [45]. Fixed collagen scaffolds were cleared in consecutive washes of 1 h, under gentle agitation in 500 µL of 28.75% *w*/*v*, 57.5% *w*/*v*, and 115% *w*/*v* D(−)-fructose (Sigma-Aldrich, Cat No. F0127) with 0.5% (*v*/*v*) α-thioglycerol in MQ. 

Cleared collagen scaffolds were cut into paper-thin sections with a scalpel. These cross-sections were either cut from the same side as the cell seeding direction or from the opposite side. CellTrace Yellow was excited with a WLL at 546 nm, and emission was collected between 570 and 600 nm using a Leica TCS SP8 SMD with an HCX PL APO 10×/0.40 or an HC PL APO CS 40×/0.85 dry objective lens and a temperature-controlled stage at 36.5 °C. Consecutive Z-stacks were acquired with some overlap to cover the entire depth of the section of the collagen scaffold and were subsequently stitched together pairwise, with linear blending and automatic computation of overlap, using the ImageJ plugin developed by Preibisch et al. [46].

### 2.8. Determination of Pre-Lyophilization Collagen Scaffold Volume

To ensure that multiple lyophilization procedures on the same collagen scaffold would not hamper cell viability or the integrity of the scaffold morphology, various loading (pre-lyophilization) volumes—100, 200, and 400 µL MQ—were investigated. The first half of the tested volume was added dropwise to one side of the collagen scaffold incubated for 15 min at 37 °C. Thereafter, the scaffold was flipped with sterile tweezers, and the remaining volume was added dropwise to the opposite side of the collagen scaffolds, followed by lyophilization overnight, as detailed in Section 2.2. Re-lyophilized collagen scaffolds were then seeded with 1.5 × 10^6^ C2C12 cells. After 24 h, the collagen scaffolds were stained with Calcein-AM and PI and imaged with an HCX PL APO 10×/0.40 or HC PL APO CS2 63×/1.20 water as outlined in Section 2.6. 

### 2.9. Dose–Response Profile of PF14-SecNLuc mRNA-Loaded Collagen Scaffolds

A concentration range from 2.5 µM to 10 µM of PF14-SecNLuc mRNA NPs (corresponding to 268.3 to 1073.2 ng of mRNA per scaffold) was loaded onto collagen scaffolds. For each concentration tested, a separate NP formulation was formed. The collagen scaffolds were loaded with 200 µL of PF14-mRNA NPs by dropwise addition of 100 µL (half) of the volume to one side of the collagen scaffold, followed by incubation for 15 min at 37 °C. After that, the scaffold was flipped with sterile tweezers. The remaining volume (100 µL) was added dropwise to the opposite side of the collagen scaffold, followed by lyophilization overnight, as detailed in Section 2.2. Subsequently, collagen scaffolds were seeded with 1.5 × 10^6^ C2C12 cells in 24-wells plates and incubated for 24 h at 37 °C with 5% CO_2_. After 24 h, the plates were gently stirred for 2 min at 100 rpm to remove any concentration gradients that might be present in the supernatant. 

The collagen scaffolds were then transferred to a new well and digested with 500 µL of 1 mg mL^−1^ collagenase from Clostridium histolyticum (Sigma-Aldrich, Cat No. C5138) in PBS for 24 h at 37 °C with 5% CO_2_. For both, the supernatants and the digested scaffolds, the extent of luciferase production was determined using the Nano-Glo Luciferase Assay (Promega, Madison, WI, USA, Cat No. N1130) according to the manufacturer’s instructions. Briefly, 50 µL of the sample was mixed with a 1:50 dilution of Nano-Glo luciferase assay substrate in Nano-Glo luciferase assay buffer. The resulting mixture was incubated at room temperature, hidden from light, for at least 3 min in a black clear flat bottom 96-wells plate (Corning Inc., Corning, NY, USA, Cat No. 3631). Importantly, an inter-sample distance in the 96-well plate of at least two columns ensured no signal crosstalk between experimental conditions. Luminescence was measured after briefly shaking the plate using the VICTOR X3 Multilabel Plate Reader (Perkin Elmer, Waltham, MA, USA).

### 2.10. Effects of Storage Temperature on Luciferase mRNA Transfections

The effect of long-term storage at different temperatures on the ability of collagen scaffolds to successfully transfect cells was explored by loading scaffolds with 5 µM PF14-SecNLuc mRNA (corresponding to 536.6 ng mRNA per scaffold) or an equivalent mRNA dose of LMM. After lyophilization, mRNA-loaded collagen scaffolds were transferred to 24-well plates, sealed, and stored for two weeks at −20 °C (freezer), 4 °C (fridge), or 21 °C (RT), all protected from light. After two weeks, collagen scaffolds were loaded with 1.5 × 10^6^ MC3T3 cells and secreted and sequestered luciferase expression was assessed as outlined in Section 2.9.

### 2.11. In Vitro BMP-7 Production in BMP-7 mRNA-Loaded Collagen Scaffolds

Collagen scaffolds were loaded with 5 µM PF14-BMP-7 mRNA (corresponding to 536.6 ng mRNA per scaffold) or an equivalent dose of LMM-formulated mRNA. BMP-7 was detected in supernatants and digested scaffolds using an in-house optimized sandwich ELISA protocol based on the R&D systems DuoSet ELISA. Nunc-Immuno MicroWell 96 well plates (Sigma-Aldrich, Cat No. M9410) were coated with 100 µL of 2 ng/µL human BMP-7 monoclonal antibody (mAb; R&D Systems Cat No. MAB3541) in PBS overnight at room temperature while sealed with Titer-Tops (Diversified Biotech, Dedham, MA, USA) and stirred at 500 rpm on an orbital shaker with a shaking stroke of 4.5 mm (IKA, Staufen, Germany). After overnight incubation, all wells were washed with PBS + 0.05% Tween-20 (Thermo Fisher Scientific, Cat No. 003005) three times with 30 s of stirring and thorough decanting by blotting the inverted plates against clean paper towels for each washing cycle. Blocking was performed with 300 µL of PBS + 5% *w*/*v* bovine serum albumin (BSA) fraction V (Roche, Basel, Switzerland, Cat No. 10735108001) in a sealed and stirred plate for 1 h. After three additional washes, 100 µL of the sample (digested scaffold in PBS + 0.1% *w*/*v* collagenase) was added and incubated in a sealed and stirred plate for 2 h. 

A standard curve was made in triplicate by twofold serial dilution of recombinant human BMP-7 (R&D Systems, Cat No. 354-CP-010/CF) spanning from 3000 pg mL^−1^ to 11.72 pg mL^−1^ in the same solution and plate as the sample (PBS + 0.1% *w*/*v* collagenase) in Protein LoBind tubes. After five washing steps, 50 ng/well of human BMP-7 biotinylated mAb (R&D Systems, Cat No. BAM354) was added and incubated in a sealed and stirred plate for 2 h. After washing five times, streptavidin-horseradish peroxidase (Strep-HRP; R&D Systems, Cat No. Dy998) 1:200 was incubated in a stirred plate for 20 min in the dark. To minimize the background signal from unbound Strep-HRP, plates were washed for seven cycles, after which 100 µL of 3,3′,5,5′-tetramethylbenzidine (TMB; Thermo Fisher Scientific, Cat No. N301) was added for maximally 20 min. The reaction was stopped by adding 50 µL 0.18 M H_2_SO_4_, followed by measuring the optical density with a Benchmark Plus Microplate Reader (Bio-Rad) at 450 nm with a wavelength correction at 540 nm.

For 2D BMP-7 mRNA transfections, 10,000 MC3T3 or C2C12 cells were seeded in 96-well plates (Greiner Cellstar, Cat No. 655180) 24 h prior to transfection. The next day, cells were transfected with 50, 100, or 200 ng of LMM-formulated BMP-7 mRNA per well. At 24 h post-transfection, cell culture supernatants were collected, and the BMP-7 ELISA was performed according to the procedures described above with the following alterations: 200 ng/well capture mAb, 25 ng/well detection mAb, and Strep-HRP diluted 1:400. The standard curves and samples were also prepared in the appropriate medium for C2C12 and MC3T3 cells.

### 2.12. Data Analysis and Statistics

Data analysis was performed using GraphPad Prism (GraphPad Software, version 8.4.2, San Diego, CA, USA). Unless stated otherwise, data are presented as means + standard error of the mean (SEM). Means were calculated by first averaging the technical replicates, for which outliers were identified using Grubbs’ test, followed by averaging the biological replicates. All data were checked for normal distribution with a Shapiro–Wilk test prior to statistical analysis. Statistical analysis was done using one- or two-way analysis of variance (ANOVA). Correction for multiple comparisons was performed using Tukey’s test with 95% confidence intervals. For the BMP-7 ELISA, the coefficient of variation was determined both in the standards and in samples and was ≤20%. The standard curve was fitted using a sigmoidal four-parameter logistic curve. The standard curve was back fitted with ±10% accuracy to verify the correctness of the fit. The signal from samples were blanked with the appropriate solution before calculating interpolations. *p* > 0.05 was considered not significant, and *p* values were reported using the GraphPad Prism style (* *p* ≤ 0.05, ** *p* ≤ 0.01, *** *p* ≤ 0.001, and **** *p* ≤ 0.0001).

## 3. Results

### 3.1. mRNA Nanoparticles Distribute Homogeneously throughout a 3D Collagen Scaffold after Lyophilization

The suitability of collagen scaffolds as lyophilizable carriers of peptide-mRNA NPs was assessed by loading 2 µM of PF14-formulated Cy5-eGFP mRNA NPs onto collagen scaffolds, incubation for 1 h, followed by freeze-drying. The PF14-Cy5-eGFP mRNA NPs were characterized by DLS and showed a monodisperse particle-size distribution (Appendix A) with an average diameter of 84.9 ± 1.6 nm (Appendix A). The collagen scaffolds were cut into slices, reswollen in 500 µL MQ, and the distribution of PF14-Cy5-eGFP mRNA NPs was assessed by confocal microscopy. Collagen scaffolds were cut either from the same (Figure 1A; cut left) or opposite direction (Figure 1B; cut right) as mRNA NP loading to ensure that sectioning of the collagen scaffolds did not introduce artifacts. 

In either case, the Cy5 signal was distributed throughout the entire imageable depth of the collagen scaffolds (Figure 1C). Moreover, the signal followed the structure of the collagen fibers, indicating that the mRNA NPs decorated the scaffold. Additionally, scaffolds were incubated for 4 h and 24 h with mRNA NPs, but this did not increase the signal or distribution (data not shown). The intensity of the Cy5 signal was quantified by plotting intensity profiles across the entire width of the scaffold. It should be noted that scaffolds were loaded with 50 µL mRNA NPs instead of 200 µL, which was used for later experiments to improve collagen structure after lyophilization. Therefore, the thickness of the scaffold was ~2 mm instead of ~4 mm. As the Cy5 signal was present throughout ~1200 µm of a 2000 µm scaffold (~60%), we decided to load the scaffolds from both sides in upcoming experiments to ensure full decoration of the collagen scaffold with mRNA NPs.

### 3.2. Cell Seeding Density and Pre-Lyophilization Volume of Collagen Scaffolds Are Crucial Parameters for In Vitro Cell Viability

After demonstrating that mRNA NPs are distributed throughout the collagen scaffold, we sought to demonstrate that the NPs yield transfection of cells seeded onto the scaffold. The initial experiments had shown that seeding density was critical in maintaining cell viability in the scaffolds. Therefore, we loaded the scaffolds with 1 × 10^5^, 5 × 10^5^, or 1 × 10^6^ C2C12 cells. After allowing cell attachment for 1 h, 500 µL of complete medium was added, and cells were grown for 48 h before staining the scaffold with calcein-AM and propidium iodide (PI). Confocal imaging revealed that both 1 × 10^5^ and 5 × 10^5^ C2C12 cells per collagen scaffold did not result in viable cells, as evidenced by the widespread occurrence of PI signal and the rounded morphology of calcein-AM-positive C2C12 cells (Appendix A). 

By comparison, 1 × 10^6^ C2C12 cells spread throughout the scaffold and maintained viability (Figure 2A). Additionally, the scaffolds were imaged at a higher resolution to gain a more detailed insight into the morphological differences between various seeding densities. This revealed that although not all cells in the 1 × 10^5^ and 5 × 10^5^ C2C12s per scaffold conditions were PI-positive, the cells did not show a normal, spread-out morphology (Appendix A). In the 1 × 10^6^ cells condition, viable cells exhibited an extended morphology along the contours of the collagen fibers (Figure 2B). 

Initially, cell viability was difficult to establish and reproduce. The suspected culprit for these inconsistencies was the pre-lyophilization volume (i.e., the added volume of mRNA NPs) of the collagen scaffolds, which sometimes led to shrinking or deformation. To this end, 100, 200, and 400 µL of MQ were loaded onto collagen scaffolds, followed by lyophilization and seeding of 1.5 × 10^6^ C2C12 cells in a volume of 100 µL. Confocal imaging of the collagen scaffolds revealed that 100 µL as a pre-lyophilization volume did not result in viable cells (Appendix A). There were no notable differences in cell viability between the 200 µL and 400 µL conditions (Appendix A). However, with a pre-lyophilization volume of 400 µL, collagen scaffolds were fully saturated, potentially leading to wasting mRNA NPs and concurrent overestimating of mRNA dose. Therefore, all subsequent experiments were performed using a pre-lyophilization volume of 200 µL. Additionally, to ensure optimal cell spreading, all scaffolds were seeded with 1.5 × 10^6^ cells in upcoming experiments.

### 3.3. C2C12 and MC3T3 Cells Decorate Collagen Fibers throughout a 3D Scaffold

Having validated the distribution of mRNA NPs throughout scaffolds and the requirements for cell viability, we sought to determine the penetration depth of cells into the scaffolds. To this end, 1.5 × 10^6^ MC3T3 or C2C12 cells were seeded in collagen scaffolds and incubated for 24 h. Following the staining of cells and optical clearing of the scaffolds, consecutive Z-stacks were acquired in sliced scaffolds to determine the presence of viable cells and their distribution throughout the 3D collagen scaffold. Collagen scaffolds seeded with C2C12 cells were cut from the same direction as cell seeding, whereas MC3T3-seeded collagen scaffolds were cut from the opposite side of cell seeding to account for potential artifacts introduced by cutting of the scaffolds. Viable and dead C2C12 (Figure 3A) and MC3T3 (Figure 3B) cells could be distinguished based on their morphology, where the small and rounded cells were presumed dead, and larger cells with a spread-out morphology were presumed viable. 

To obtain a semi-quantitative assessment of penetration depth, the intensity of the CellTrace signal was plotted along the entire height and width of the stitched image. C2C12 cells reached as far as ~1500 µm into the 3D collagen (Figure 3C) scaffold 24 h post-seeding, and MC3T3 cells penetrated as deep as ~1750 µm (Figure 3D). Both cell types attached to the collagen fibers. Regardless of cell type, there was a cell density gradient from the surface into the interior of the scaffold. 

### 3.4. Peptide-Mediated mRNA Delivery in 3D Collagen Scaffolds Induces Dose-Dependent Protein Production

Previously, we have demonstrated that mRNA dose correlates linearly with protein expression over several orders of magnitude in both conventional 2D cell culture experiments and mice [47]. Next, we wanted to learn whether a dose–response function could be observed for the mRNA-functionalized collagen scaffolds. Scaffolds were loaded with increasing amounts of PF14-formulated SecNLuc mRNA nanoparticles, which had a similar size and monodispersity as the Cy5-eGFP mRNA nanoparticles (Appendix A). 

Having demonstrated that both nanoparticles and cells are present throughout the scaffolds, we determined luciferase concentrations (Figure 4A) in the supernatants (Figure 4B) as well as in the collagenase-digested scaffolds (Figure 4C). The amount of sequestered protein was substantially higher than the one of secreted protein in all conditions (Figure 4C). A doubling of mRNA dose from 268.3 ng to 536.6 ng mRNA per scaffold resulted in a four-fold increase in total luciferase detected. In contrast, from 536.6 ng to 804.9 ng mRNA, the amount of luciferase only increased by 30% and leveled out when increasing the amount to 1073.2 ng mRNA per scaffold (Figure 4D).

Since C2C12 cells have been reported to be hard-to-transfect [48,49], we sought to compare the transfection efficiencies of these cells with those of MC3T3 cells (Appendix A). Using identical experimental conditions, peptide-mediated SecNLuc mRNA transfection of MC3T3 cells yielded significantly higher (*p* < 0.0001) luciferase expression. SecNLuc-mRNA-loaded scaffolds seeded with MC3T3 cells yielded 4.26 times more luciferase expression than those loaded with C2C12 cells. Accordingly, MC3T3 cells were used for all subsequent experiments.

### 3.5. Long-Term Storage of mRNA-Loaded Collagen Scaffolds at Different Temperatures Decreases Transfection Efficiencies

Continuity of the cold chain was a major concern for the mRNA SARS-CoV2 vaccines. Even though the initial requirement for freezing at −80 °C was later changed to −20 °C, storage at above freezing temperature would considerably increase the ease of handling. Therefore, we sought to assess the long-term stability of lyophilized collagen scaffolds loaded with mRNA NPs at different temperatures. LMM-formulated mRNA transfections have previously been shown to be compatible with lyophilization [50]. Moreover, since the long-term stability of LNP-formulated mRNA is well-established [51,52,53], we included LMM mRNA NPs as a control for LNP storage stability.

To this end, scaffolds were seeded with PF14- or LMM- formulated SecNLuc mRNA, with mock-loaded scaffolds serving as untreated controls, and all stored for two weeks at −20 °C, 4 °C, or 21 °C. After storage, MC3T3 cells were seeded, and luciferase expression was quantified. For storage of collagen scaffolds at −20 °C for two weeks, LMM-formulated mRNA significantly outperformed scaffolds loaded with PF14-formulated mRNA (Figure 5A). The secreted luciferase signal by PF14-formulated mRNA was about two orders of magnitude higher than the one of the untreated condition. In contrast, with scaffold storage at 4 °C, PF14 significantly outperformed the LMM-formulated mRNA (Figure 5B). For storage at 21 °C, both PF14- and LMM-formulated mRNA scaffolds yielded significantly higher luminescence values than untreated scaffolds, but without significant differences between LMM and PF14 (Figure 5C). Overall, the signal decreased by a factor of about 10 for increasing storage temperatures.

Comparing the transfection efficiencies of PF14-formulated mRNA NPs of storage conditions versus freshly prepared scaffolds revealed to which extent luciferase expression diminished (Appendix A). The unstored scaffolds gave rise to significantly more (*p* = 0.0103) luciferase expression than scaffolds stored at −20 °C, with unstored scaffolds producing 12 times more luciferase on average. With storage at 4 °C, differences were further exacerbated, where unstored scaffolds produced 66 times more luciferase (*p* = 0.0076). Overall, these results indicate that despite not being specifically formulated for long-term storage, peptide-mRNA NPs outperform LMM NPs in terms of long-term storage at 4 °C.

### 3.6. BMP-7 mRNA Transfection in Collagen Scaffolds Results in Biomaterial-Mediated Protein Production

After establishing reporter gene expression, we finally probed for the expression of BMP-7 as a therapeutically relevant protein. Using conventional 2D cell culture experiments, transfection with LMM-formulated BMP-7 mRNA showed an excellent dose–response function in both cell types, where all doses were significantly different (*p* ≤ 0.0030 for all conditions) from each other within the same cell type (Figure 6A).

Next, collagen scaffolds were loaded with 536 ng BMP-7 mRNA per collagen scaffold. Again, PF14 and LMM-formulated mRNA were tested (Appendix A). In line with previous observations, LMM mRNA NPs were substantially larger (~9-fold) than PF14 mRNA NPs [44]. After 48 h, BMP-7 expression could readily be detected in the scaffolds after digestion (Figure 6B). However, BMP-7 was not released in detectable levels into the supernatant 24 h post-transfection. Despite similar expression levels for C2C12 and MC3T3 cells with LMM-formulated BMP-7 mRNA in 2D transfections, C2C12 conditions did not show any significant BMP-7 production (data not shown). Considering that MC3T3 cells outperformed C2C12 cells by a factor of about 4 with SecNLuc transfections (Appendix A), the significant difference between PF14 conditions (*p* < 0.0001) follows expectations.

Accounting for the difference in BMP-7 detection between 2D and 3D, and rather short half-life of BMP-7, the reduction of produced protein in 3D collagen scaffolds versus 2D by ~4- to ~30-fold seems reasonable. In MC3T3 cells, the PF14 formulation was significantly different from the untreated condition (*p*: 0.0026) whereas the LMM formulation was not (Figure 6B). 

## 4. Discussion

Engineered biomaterials for in situ bone regeneration represent a promising and rapidly growing approach for treating poorly healing bone defects. Here, we demonstrate the incorporation of peptide-formulated mRNA into porous collagen scaffolds yielding nanoparticle-functionalized collagen-based biomaterials that can be lyophilized, stored and achieve transfection of cells in a dose-dependent manner. 

The initial characterization of the collagen scaffolds revealed that mRNA NPs dispersed throughout the majority of the scaffold, even after lyophilization. We did not explicitly address the molecular nature of the interaction by which the polyplexes interact with the collagen. At an N/P ratio of 3, PF14 mRNA polyplexes have a positive zeta potential. However, a slightly positive zeta potential has also been reported for collagen fibrils, suggesting that hydrophobic interactions may contribute to the interaction [54]. A systematic investigation of the interdependence of charge and loading capacity would be problematic as, at lower N/P ratios, particles tend to aggregate, and at higher N/P ratios, solutions contain an excess of peptide that is not part of the polyplexes.

Interestingly, in our case, higher cell densities by about a factor of 10 were needed in order to obtain viable cells compared to previous reports in a study seeding NIH3T3 cells onto equine-derived collagen scaffolds [33]. A potential explanation may be found in this earlier study’s rather clustered cell growth. In contrast, in our case, the two different cell types were distributed homogenously throughout the major part of the scaffold.

For transfections with SecNLuc and BMP-7 mRNA, most of the protein produced was locally sequestered and only released upon enzymatic digestion of the collagen scaffolds. This retention of proteins in the scaffold aligns with the goal of avoiding systemic exposure when used in vivo. Messenger RNA will yield expression over several days, which may be further extended by sustained mobilization of mRNA nanoparticles from the scaffold and uptake into cells. At this point, we cannot demonstrate whether the retention of proteins was due to slow diffusion or to interactions with the collagen scaffold, which may very well be the case for BMP-7 [55]. Many components of the extracellular matrix possess growth factor binding sites.

Moreover, it should be noted that the differences between the 2D and 3D experimental setups for BMP-7 mRNA transfections were substantial. The 3D experiment in collagen scaffolds was seeded with 150 times the amount of cells, while the mRNA dose was only scaled up by a factor of ~11. When dividing the mRNA dose by the initial amount of seeded cells, this yields 20, 10 and 5 pg of BMP-7 mRNA per cell in 2D, for doses of 200, 100 and 50 ng mRNA, respectively. For the 3D collagen scaffolds, the mRNA dose would be 0.4 pg/cell. These mRNA dose differences, most likely explain the differences in protein expression. 

Lyophilized collagen scaffolds are robust biomaterials that can be stored at ambient temperature for months [54]. It would be highly desirable if this characteristic also applied to functionalizations incorporated into the scaffolds. Unless precautions are taken, RNases are omnipresent, and already a single change such as a base oxidation or a strand break in the mRNA can inactivate an mRNA molecule [7]. Complexation of mRNA with lipid and peptide carriers has been shown to enhance stability and reduce nuclease susceptibility [56,57,58]. There was a reduction of activity by a factor of about 10 when scaffolds were stored at 4 °C in comparison to storage at −20 °C for PF14-formulated mRNA and even more for LMM-formulated mRNA. The significantly better preservation of activity at 4 °C for the PF14 formulation in comparison to the LMM formulation is in line with previous observations that PF14 formulations show more robustness in complex environments [41]. Storage at ambient temperature reduced activity to around background levels for both formulations. We should stress that no specific measures were taken to ensure the absence of RNase activity for the collagen scaffolds. Future efforts should clarify whether stability and transfection efficiency can be improved through washing steps to remove potential RNase activity or through the addition of cryoprotectants. For instance, lipid mRNA NPs have been shown to benefit from the formulation with mannitol, lactose, trehalose, and sucrose [51,52]. For a lipid nanoparticle-based formulation, embedded within a collagen scaffold of equine origin, a greatly extended half-life was observed [34], which may be due to vacuum sealing of the scaffolds.

Previously, we have sought to mathematically model relevant differences between PF14 and LMM mRNA NPs [47]. With LMM mRNA NPs being roughly 10-fold larger than PF14 NPs, they should contain about 100 times as many mRNA molecules per particle. Therefore, when using equal mRNA doses, the collagen scaffolds could be decorated with ten times fewer NPs, which could explain the significantly better performance of our PF14-based mRNA formulations inside the 3D collagen scaffolds by virtue of more homogeneous NP loading.

PF14-formulated mRNA NPs have several advantages over lipid-formulated NPs. The formulation requires only one component, and nanoparticles of a similar size and uniformity as LNPs can be obtained through simple mixing of aqueous solutions. For the same reason, other polymer-based formulation strategies next to LNPs are still being explored, also for biomaterials [59]. A unique characteristic of the PF14-based formulation is the composition of natural building blocks, potentially resulting in a truly traceless mRNA transfection method. Moreover, PF14 contains four residues of the non-proteinogenic amino acid ornithine. The replacement of lysine residues with ornithine has been shown to increase the in vitro transfection of plasmid DNA transfection complexes up to 10-fold due to formation of more stable complexes [60]. Additional benefits of ornithine substitution include enhanced protease resistance and increased polyplex stability [61,62]. LNP formulations are confronted with the need for PEG as a shielding agent, which introduces a non-degradable component and can lead to unwanted adverse reactions [63]. Therefore, the field has been looking for alternatives [64,65]. Moreover, impurities of lipid-based mRNA NPs can result in adduct formation through the covalent addition of reactive lipid species to the nucleobase, resulting in decreased mRNA activity and consequently lower protein expression [66].

## 5. Conclusions

Collectively, we demonstrate the generation and thorough characterization of collagen scaffolds functionalized with a peptide-based mRNA formulation. Importantly, we show that our approach yields a homogenous distribution of the mRNA nanoparticles throughout the scaffolds. Moreover, successful mRNA transfection leads to local sequestration of produced protein. Additionally, our peptide-mRNA NPs significantly outperformed the commercially available LMM NPs in long-term storage experiments at 4 °C. Likewise, mRNA transfections with PF14 in scaffolds gave rise to more BMP-7 production. Further in vivo studies are required to demonstrate the translational potential of these off-the-shelf mRNA-loaded collagen scaffolds for bone regeneration applications.

## Figures and Tables

**Figure 1 pharmaceutics-14-01619-f001:**
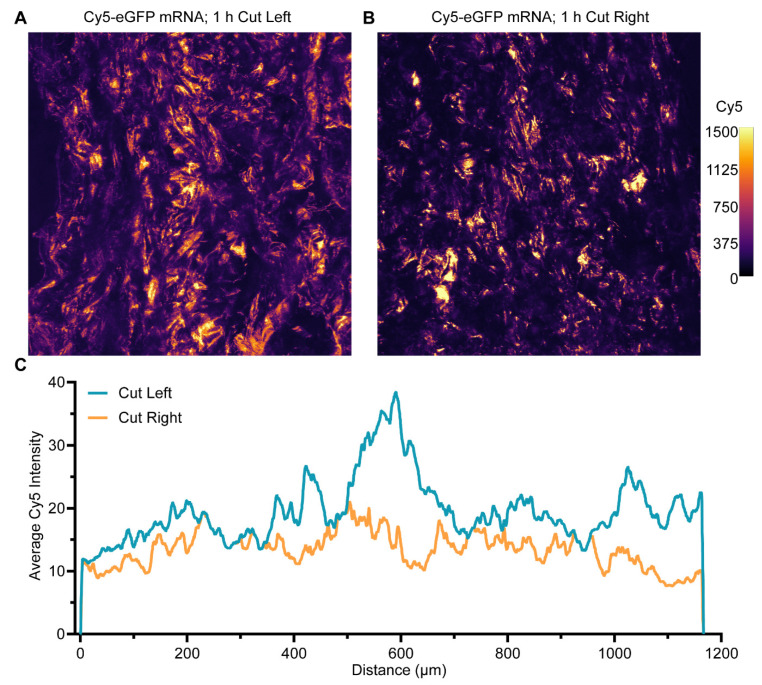
Distribution of peptide-mRNA nanoparticles throughout three-dimensional collagen scaffolds post-lyophilization. PF14-Cy5-eGFP mRNA NPs (214 ng mRNA/scaffold) were loaded onto collagen scaffolds and allowed to diffuse for 1 h, followed by lyophilization, re-swelling, and cutting of orthogonal slices. (**A**) Average Z-projection of 5 confocal sections spanning the entire depth of the cut slice of the scaffold, cutting direction is equal to mRNA NP loading direction. (**B**) Average Z-projection of 3 confocal sections spanning the entire depth of the cut slice of the scaffold, cutting direction is opposite to mRNA NP loading direction. Cy5 intensities are represented by a false-color look-up table (right), and brightness and contrast were equally adjusted across conditions. (**C**) Average Cy5 intensity profiles were measured in Z-projections of (**A**; blue) and (**B**; orange).

**Figure 2 pharmaceutics-14-01619-f002:**
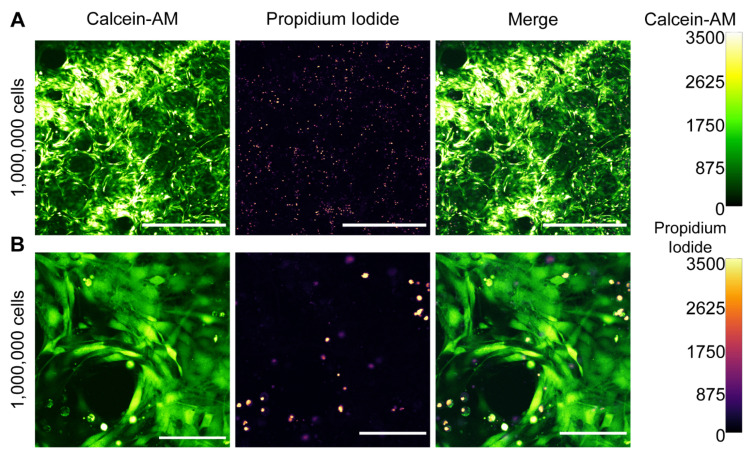
Sufficient initial cell densities are needed for C2C12 cell viability and spreading in 3D collagen scaffolds. (**A**) Low magnification overview of C2C12 viability 48 h post-seeding; (**B**) high magnification zoom-in. Calcein-AM intensities to assess cell viability and PI staining are visualized by the false color look-up tables green hot and mpl-inferno, respectively (right). Brightness and contrast were equally adjusted across conditions. Scale bars in panel A represent 500 µm, scale bars in panel B represent 100 µm.

**Figure 3 pharmaceutics-14-01619-f003:**
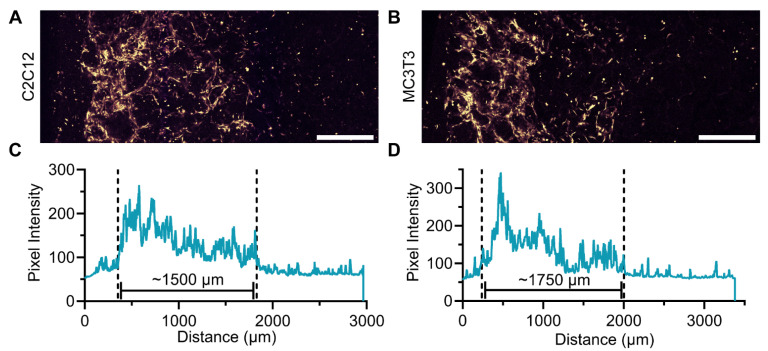
Cell penetration into the 3D collagen scaffolds. Collagen scaffolds were seeded with 1.5 × 10^6^ cells and after 24 h stained with CellTrace Yellow, fixed, and cleared. (**A**) Stitched image of 3 neighboring Z-stacks of C2C12 cells, cutting direction is equal to cell seeding direction. (**B**) Stitched image of 3 neighboring Z-stacks of MC3T3 cells, cutting direction is opposite to cell seeding direction. (**C**,**D**) Cy5 intensity profiles measured in Z-projections of (**A**) C2C12 cells and (**B**) MC3T3 cells, a pixel intensity threshold of 100 was used to quantify depth of cell penetration throughout the collagen scaffold. Brightness and contrast were equally adjusted across conditions. Scale bars represent 500 µm.

**Figure 4 pharmaceutics-14-01619-f004:**
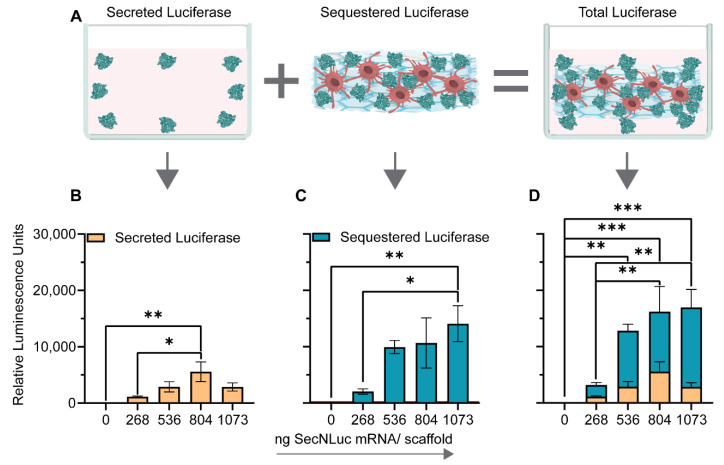
Dose–response function of C2C12 cells seeded on 3D collagen scaffolds functionalized with PF14-formulated secreted nanoluciferase mRNA nanoparticles. (**A**) Schematic of the experimental detection strategy. Secreted nanoluciferase was detected 24 h post-transfection, followed by collagenase digestion of the scaffold for 24 h, in which the amount of sequestered nanoluciferase was determined. (**B**) Quantification of secreted and (**C**) sequestered nanoluciferase. (**D**) Merged quantification of secreted and sequestered nanoluciferase by chemiluminescence. Data represent the mean + SEM of two independent experiments. *: *p* ≤ 0.05, **: *p* ≤ 0.01, ***: *p* ≤ 0.001.

**Figure 5 pharmaceutics-14-01619-f005:**
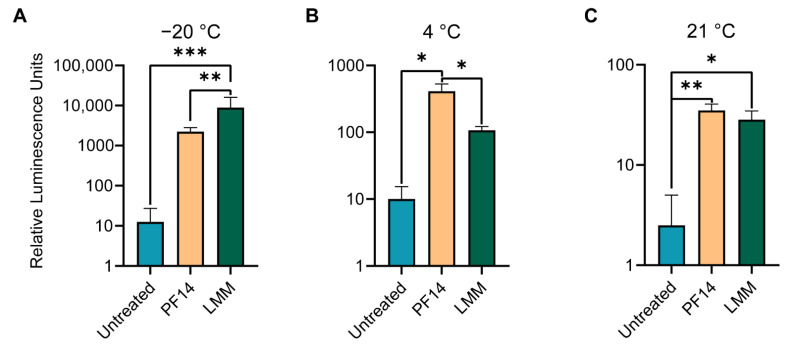
The effects of storage temperature on transfection of mRNA-functionalized collagen scaffolds. MC3T3 cells were seeded onto scaffolds containing PF14-SecNLuc mRNA (536 ng mRNA/scaffold), or LMM-formulated mRNA at equivalent dose. Before seeding, all collagen scaffolds had been stored at the indicated temperatures for 2 weeks in a sealed 24-well plate, protected from light. (**A**) Merged quantification of secreted and sequestered nanoluciferase in 3D collagen scaffolds stored for 2 weeks at −20 °C, (**B**) 4 °C, (**C**) and at 21 °C. Values represent the mean + SEM of three biological replicates (two for untreated conditions), *: *p* ≤ 0.05, **: *p* ≤ 0.01, ***: *p* ≤ 0.001. PF14: PepFect14, LMM: Lipofectamine MessengerMAX.

**Figure 6 pharmaceutics-14-01619-f006:**
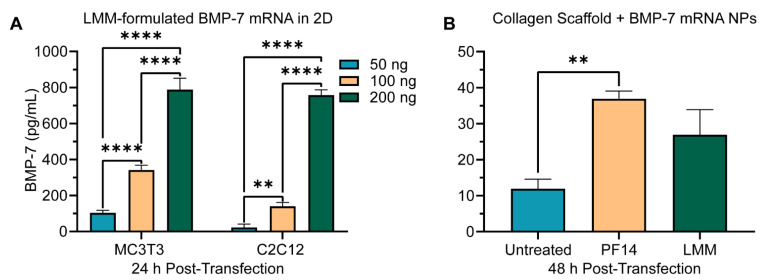
Transfection of cells with BMP-7 mRNA formulations. (**A**) Dose–response function of BMP-7 mRNA transfections with LMM in 2D cell culture. Data represent the mean + SEM of three biological replicates of BMP-7, secreted in the supernatant 24 h post-transfection. (**B**) MC3T3 cells were seeded in collagen scaffolds loaded with 5 µM PF14 BMP-7 mRNA (536 ng mRNA/scaffold) nanoparticles, or LMM formulated mRNAs at equivalent dose. BMP-7 expression was determined by ELISA after enzymatic digestion of the collagen scaffold. Data represent the mean + SEM of two biological replicates of the sequestered BMP-7 protein. **: *p* ≤ 0.01, ****: *p* ≤ 0.0001. PF14: PepFect14, LMM: Lipofectamine MessengerMAX.

## Data Availability

The data presented in this study are available on request from the corresponding author.

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
