# Peer review of "Biomaterial-Mediated Protein Expression Induced by Peptide-mRNA Nanoparticles Embedded in Lyophilized Collagen Scaffolds"

_pharmaceutics, 2022, doi:10.3390/pharmaceutics14081619_

Round 1
Reviewer 1 Report
This work investigates the use of cell-penetrating peptides to aid localized mRNA uptake by cells seeded on collagen scaffolds, and their stability during storage. It is a well conceived and implemented piece of research that is clearly described in the manuscript with potential biomedical applications in bone regeneration.
Reviewer 2 Report
R. Brock and his coworkers demonstrated that the polyplex of cationic cell-penetrating peptide PepFect14 (PF14) and mRNA is successfully loaded and uniformly distributed into a porous collagen scaffold. This collagen scaffold preserved the cell binding and viability and successfully produced the proteins; bone morphogenetic protein-7, and luciferase. Indeed, with a battery of well-done experiments, the authors demonstrated the advantage of collagen scaffold in retaining the activity at -20 ℃, 4 ℃, and 21 ℃ over two weeks. Overall, the manuscript was well-constructed. This is an exciting manuscript suitable for publication in MDPI Pharmaceutics. However, I recommend a detailed revision addressing the following minor issues carefully to reach out to more audiences and readers of different disciplines.
1. Define the one-letter codes of Stearyl-AGYLLGKLLOOLAAAALOOLL-NH2 (PepFect14) after its first appearance in the text.
ïƒ Stearyl-Ala-Gly-Tyr-Leu-Leu-Gly-Lys-Leu-Leu-Orn-Orn-Leu-Ala-Ala-Ala-Ala-Leu-Orn-Orn-Leu-Leu- NH2.
2. It is good to show the chemical structure of PF14, at least in the Supporting Information.
3. I recommend the authors furnish a Scheme representing the whole theme of the work.
4. It is recommended to discuss the role of non-proteogenic amino acid ornithine in PepFect14? From the amino acid sequence of PF14, a modified version of stearyl-transportan10, it is observed that the primary cationic acid is ornithine (4 residues) and lysine (1 residue). The primary cationic amino acid in stearyl-TP10 is Lysine (4 residues). For example, replacing lysine with ornithine confers resistance against peptidase and protease degradation(Nucleic Acids Res. 2011;39(12):5284-98). Furthermore, mRNA complexation with poly(L-ornithine) showed higher resistance against ribonucleases than poly(L-lysine), thus resulting in higher protein production efficiency (Macromol Rapid Commun. 2022;43(12):e2100754)
5. What is the driving force of PF14-mRNA polyplex loading and binding in porous collagen scaffolds? What is the zeta-potential of PF14-BMP-7 mRNA polyplex? Is there any relation between loading into collagen scaffolds and the surface charge of PF14-mRNA polyplex?
Reviewer 3 Report
Authors have written a research article on”Biomaterial-Mediated Protein Expression Induced by Peptide-mRNA Nanoparticles Embedded in Lyophilized Collagen Scaffolds”. The manuscript has been written well. The authors have explained every part very nicely. The manuscript can be accepted for publication after addressing the following comment.
Minor comments
1. Please correct the English of these sentences, “In our aging society, the number of patients suffering from poorly healing bone defects increases. The clinic uses the delivery of bone morphogenetic proteins (BMPs) to promote bone regeneration”. Line no 17 and 18.
2. Please provide the space between “regenerativemedicine”. Line no 30.
3. Please rephrase the sentences to get the clear meaning “For bone regeneration, in particular, such biomaterials can, on the one hand, provide a structural framework to facilitate host cell migration and attachment, and on the other hand, accelerate cell growth and differentiation”. Line no 63 and 64
4. Please replace “fluorescently” with “fluorescent” in line no 99.
Round 2
Reviewer 2 Report
The manuscript by R. Brock and his co-researchers can be accepted in the present form.